

# Photo-Induced Shrinking of Aqueous Glycine Aerosol Droplets

Shinnosuke Ishizuka,[a,b,c ‡, *] Oliver Reich,[a, ‡] Grégory David,[a] and Ruth Signorell[a]*

[a] Laboratory of Physical Chemistry, ETH Zurich, Vladimir-Prelog-Weg 2, CH-8093, Zurich, Switzerland

[b] Institute of Advanced Research, Nagoya University, Nagoya 046-8601, Japan

[c] Institute of Space-Earth Environmental Research, Nagoya University, Nagoya 046-8601, Japan

**ABSTRACT:** Due to their small size, micrometer and submicron sized solution droplets can respond differently to physical and chemical processes compared with extended bulk material. Using optically trapped micrometer sized aqueous glycine droplets, we demonstrate photo-induced degradation of glycine upon irradiation with visible light, even though molecular glycine does not absorb light in the near UV/vis range to any significant extent. This reaction is observed as photo-induced shrinking of the droplet, which we characterize by analyzing the elastic light scattering and the Raman spectrum of the droplet over the course of the reaction. We find the volume to shrink with a constant rate over the major part of the shrinking process. This indicates the presence of a rate limiting photo-catalyst, which we attribute to mesoscopic glycine clusters in the droplet solution. Our findings relate to previous reports of visible light absorption by photosensitizers. However, to the best of our knowledge, this is the first experimental evidence of a photochemical pathway facilitated by mesoscopic clusters. Light interaction with such mesoscopic photoactive molecular aggregates might be more important for aerosol photochemistry than previously anticipated.

* rsignorell@ethz.ch, ishizuka.shinnosuke@i.mbox.nagoya-u.ac.jp

‡ S.I. and O.R. contributed equally to this work.




## 1. INTRODUCTION

Aerosols, dispersions of solid and liquid particles in a gas, are ubiquitous in Earth's atmosphere and as such play an important role for many atmospheric processes (Boucher et al., 2013; Pöschl and Shiraiwa, 2015). Size, chemical composition, viscosity and thermodynamic phase of aerosol particles respond to their environment including the surrounding gas species, temperature, humidity (Bones et al., 2012; Zieger et al., 2017; Tang et al., 1997; Swietlicki et al., 2008) and light irradiation (Pöschl and Shiraiwa, 2015; Cremer et al., 2016; Walser et al., 2007; Corral Arroyo et al., 2022). Particular attention has been paid to their chemistry distinct from that in the bulk, a phenomenon possibly arising from high surface to volume ratio of aerosol particles and the accessibility to highly supersaturated states (Altaf et al., 2016; Kucinski et al., 2019; Bzdek and Reid, 2017). Various chemical reactions have been shown to be accelerated in microdroplets (Cremer et al., 2016; Lee et al., 2015; Girod et al., 2011), with some reactions being exclusive to the droplet phase (Lee et al., 2019). These unique reactors can act as medium for the birth, growth and degradation of atmospherically relevant particles (Ruiz-Lopez et al., 2020), and be utilized for organic synthesis (Bain et al., 2017). Chemical processes in prebiotic aerosols have also been proposed as potential mechanisms for the origin of life (Tervahattu et al., 2004). However, molecular processes leading to the anomalous chemistry in micrometer and submicron aerosol particles are still largely unknown. Although some processes may be ascribed to the discontinuous and asymmetric intermolecular interactions at the particle surface (Ruiz-Lopez et al., 2020), the microphysical origins behind many of the aforementioned particle specific phenomena are still not adequately explored.

Glycine is an amino acid that acts as precursor to proteins and fulfills a number of other biological functions (Arnstein, 1954; Hall, 1998; Jackson, 1991). With its small size and simple structure, glycine often serves as a proxy for other amino acids and physiologically relevant molecules, and as such has been studied extensively in the past. It is generally accepted that molecular glycine does not absorb light in the near UV/vis range, similar to other amino acids (Bhat and Dharmaprakash, 2002). However, it has been shown that their optical properties change when glycine molecules arrange themselves into mesoscopic clusters (Terpugov et al., 2021). Furthermore, these formations respond to light irradiation in non-trivial ways which can be exploited to induce long range order inside the glycine solution (Alexander and Camp, 2019; Sugiyama et al., 2012; Zaccaro et al., 2001; Garetz et al., 2002) on a scale of up to millimeters (Yuyama et al., 2010). While these interactions have the potential to change the optical properties of glycine ensembles significantly and to enable new photochemical reaction pathways, there has been little experimental evidence for such reactions so far.

In this work, we study the response of aqueous glycine droplets to irradiation by visible light. We observe the shrinking of optically trapped micrometer sized glycine droplets, which can be unambiguously attributed to the exposure to the trapping laser with wavelength 532 nm. To the best of our knowledge, this interaction has not been reported before. We characterize it here with particular focus on the shrinking rate and its dependence on the light intensity. To explain our results, we discuss possible reaction schemes based on the available experimental





data. Although further data is needed to elucidate the exact photochemical pathways of the observed reaction,
these findings demonstrate the existence of a photochemical reaction for molecules which previously have been
considered photochemically inert at visible wavelengths.






## 2. METHODS


Dual beam optical traps are widely used to confine and isolate single particles (Ashkin, 1997; Gong et al., 2018;
Čižmár et al., 2005; Esat et al., 2018; Reich et al., 2020). The counter-propagating tweezers (CPT) setup for
trapping aqueous glycine droplets is shown in Fig. 1 and consists of a continuous green laser beam (Novanta
Photonics Opus 532 6W), which is expanded and then split into two beams of equal power. These two beams are
aligned counter-propagating on a single axis and focused into the trapping cell, where a single droplet is trapped
between the two focii.

The droplets are generated from 1.0 M or 2.0 M aqueous solutions of glycine (purity ≥ 99%, Sigma-Aldrich
G7126) using a commercial atomizer (TSI 3076) with pressurized, humidified nitrogen gas (purity 5.0). A system
of copper tubings directs the spray of particles into the trapping cell, where the droplets agglomerate at the desig-
nated trapping position. The humidification of the nitrogen flow is necessary to ensure that the droplets reach the
trapping position in the liquid state. The trapping cell is filled with nitrogen gas (purity 5.0) and a steady nitrogen
flow formed by combining wet and dry nitrogen with adjustable flow ratios is used to control the relative humidity
(RH) in the cell. For the experiments reported here, the RH is set at $77 \pm 3$ % well above the efflorescence RH of
glycine at approximately 55 % (Chan et al., 2005). At this RH, the droplet solution is supersaturated with an
estimated glycine concentration of 60 % in mass (Chan et al., 2005), corresponding to approximately 5 M. Tem-
perature and RH inside the trapping cell are monitored by a sensor (Sensirion SHT35) placed a few millimeters
away from the trapping position. When the agglomerated particle reaches a size of approximately 2-3μm in radius,
the remainder of the droplets in the cell are flushed out with nitrogen for 20-30 minutes to ensure that only the
trapped droplet remains in the cell. After flushing, the power of the trapping laser is kept constant until the end of
the measurement.

The particle shrinking is monitored by imaging the polarization resolved two-dimensional angular optical scatter-
ing (polarization resolved TAOS) of the particle (Parmentier et al., 2022), as shown in Fig. 2. The TAOS image
is obtained by collecting the elastically scattered light of the trapping beams under a scattering angle of $90 \pm 24°$
with an objective (Mitutoyo 20x NA 0.42). The parallel and perpendicular polarization components with respect
to the scattering plane (TAOS PPol and TAOS SPol) are separated using a polarization beam splitter and recorded
with separate CMOS cameras (Thorlabs DCC1545M). The scattering intensity for each polarization is calculated
from the average of the respective TAOS image and recorded over time. At specific times, the shrinking spherical
particle reaches a size at which it is in resonance with the light of the trapping beams, which corresponds to a Mie
resonance (Bohren and Huffman, 2008). From the comparison of the recorded evolution of the polarization re-
solved scattering intensity to simulations using Mie theory, the size of the particle can be determined at the specific
times. Fig. 3 shows an example of such a TAOS analysis. The values of the size at the discrete points in time,
obtained from the times where the particles experience a Mie resonance, can then be interpolated with high accu-
racy to obtain the full size evolution of the particle over the course of the measurement.






The molecular composition of the particle is monitored by continuous recording of Raman spectra (David et al.,
2020) during the shrinking process. To this end, the light scattered by the particle is collected under a scattering
angle of $90 \pm 24°$ by a second objective and fiber coupled into a low noise, high sensitivity spectrograph (Andor
KY-328i-A). The inelastically scattered light is analyzed in the range 540-680 nm which corresponds to Raman
shifts of 280-4100 $cm^{-1}$. This range contains in particular the O-H symmetric stretching mode of water ($\nu_2$-$H_2O$,
2700-3750 $cm^{-1}$) as well as several vibrational modes of glycine, which we exploit for the characterization of the
molecular composition (see later data for an example).

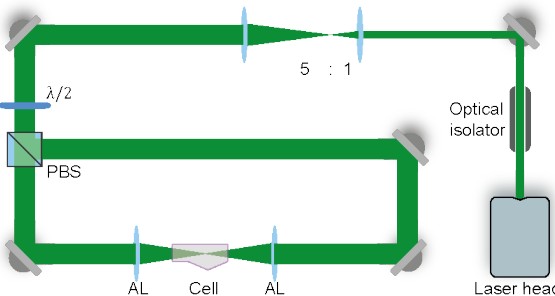


**Figure 1** Counter-propagating tweezers setup. The laser beam is first expanded by a factor of 5 and then split into
two beams by the polarizing beam splitter (PBS). The half-waveplate ($\lambda$/2) rotates the polarization to 45° with
respect to the axes of the PBS to ensure equal power splitting between the two beams. The beams are aligned on
a single axis and focused into the trapping cell using two aspherical lenses (AL). An optical isolator introduced
at the start of the beam path prevents unwanted optical feedback into the laser head.

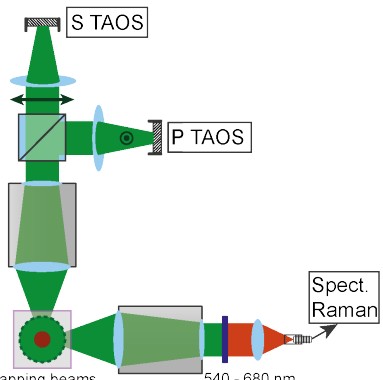


**Figure 2** Setup for TAOS imaging and Raman spectroscopy. The trapping beams run perpendicular to the figure
plane. The scattered light of the trapping beams is collected horizontally and vertically at a scattering angle of 90
$\pm 24°$. The vertical beam is split into parallel and perpendicular polarized light with respect to the scattering plane





and the respective beam is loosely focused on a CMOS camera (P TAOS and S TAOS respectively). The hori-
zontal beam is filtered for the spectral range of 540 – 680 nm and fiber coupled into a low noise, high sensitivity
spectrometer (Spect. Raman) for measurement of the Raman spectrum.

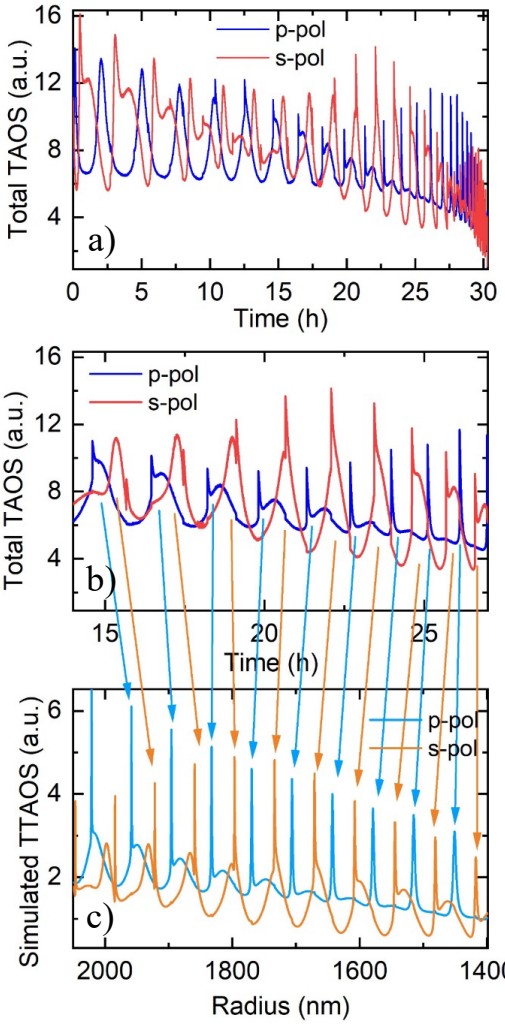


**Figure 3** Analysis of the two-dimensional angular optical scattering spectrum (TAOS). **a)** Experimental polari-
zation resolved total TAOS signal over time. **b)** Zoom on a specific time interval for clarity **c)** Simulated polari-
zation resolved total TAOS signal as function of particle radius in the specific time interval. Arrows indicate the
peak assignment based on the similarities between the peak shapes.



## 3. RESULTS AND DISCUSSION

The shrinking of the aqueous glycine droplets over time is shown in Fig. 4 for three representative examples. The droplets are trapped using different laser powers, which affects the rate at which their volume is decreasing. For all laser powers, the volume is observed to shrink linearly with time over a large portion of the shrinking process. After a certain point ('point of acceleration'), when the droplet has lost 60 – 80 % of its volume, the shrinking suddenly accelerates, only to continue again approximately linearly with time, albeit at a higher rate.

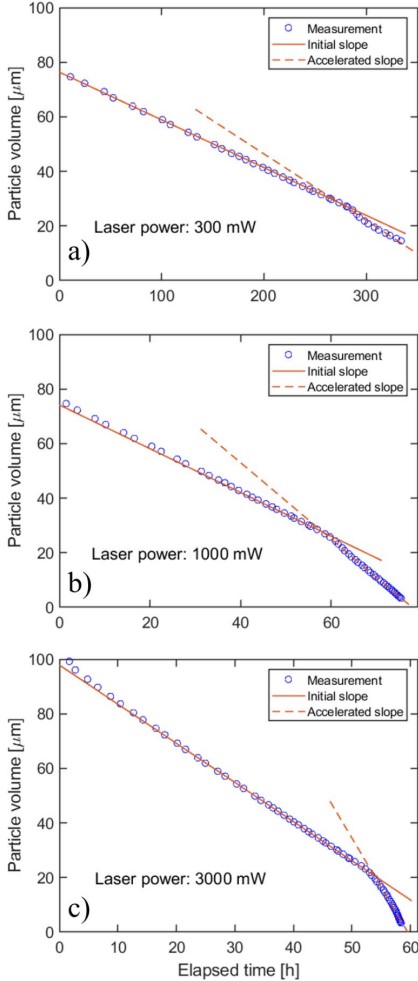

**Figure 4** Volume shrinking of glycine droplets as a function of time. **a)** Droplet trapped at 300 mW nominal laser power **b)** droplet trapped at 1000 mW nominal laser power **c)** droplet trapped at 3000 mW nominal laser power. At higher power, the shrinking proceeds faster (visible by the lager slopes). Solid and dashed lines indicate the best linear fit before and after the acceleration of the shrinking observed at approximately **a)** 280 h, **b)** 59 h and **c)** 53 h.





To quantify the dependence of the shrinking rate on the light intensity incident on the particle, a linear fit is
performed on the data before and after the point of acceleration. Since between different experiments, the align-
ment of the optical trap, and hence the focusing of the laser light on the particle, is subject to temporal mechanical
drifts, the nominal laser power used for trapping of the droplets is not an optimal indicator for the incident inten-
sity. Instead, we use the intensity of the scattered light as a measure that is proportional to the incident light
intensity. The intensity of the scattered light is obtained from the average signal of the TAOS PPol and SPol
images during the time of the droplet shrinking. To ensure consistency between the different experiments, the
average of the light intensity is taken over the same volume interval of [65.4, 51.0] $\mu m^3$ (radius [2.5, 2.1] $\mu m$) for
all droplets. This interval corresponds to the interval for which we obtained data for most droplets. Fig. 5 shows
the resulting initial shrinking rates as function of light intensity. The shrinking rate is observed to be proportional
to the light intensity, confirming that the shrinking is induced by the trapping laser at a wavelength of 532 nm.

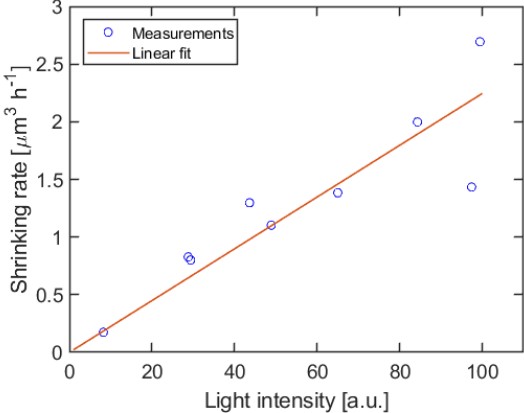


**Figure 5** Fitted shrinking rate as function of light intensity. The solid line indicates the best linear fit through
the origin.
To gain further insight into the shrinking mechanism, we analyze the evolution of the Raman spectra over the
course of the shrinking process. A representative example of such an evolution is shown in Fig. 6. For Raman
spectra of single spherical particles, so called morphology dependent resonances, or whispering gallery modes
(WGMs) (Oraevsky, 2002), are superimposed on the molecular signals. To separate the WGMs from the molecular
signal of interest, the Raman spectra are normalized and stacked in chronological order from left to right, as shown
in Fig. 6a. The WGMs show up as a manifold of thin slanted lines bending towards lower wavenumbers with
increasing time as the droplet shrinks. Molecular band positions on the other hand are independent of particle
size, and are identified as horizontal lines in the evolution of the Raman spectra.

From Fig. 6a it is evident that the molecular Raman signal remains qualitatively the same, indicating that no
significant change in the molecular composition takes place in the droplet over the course of the shrinking process.
This behavior is observed for all investigated droplets. Since the particle loses the major part of its volume during





the shrinking, this implies that glycine is removed from the droplet as a consequence of chemical reaction (*vide*
*infra*). As the water vapor pressure of the droplet is given by the surrounding RH of 77 % and therefore has to
remain constant, the removal of glycine from the droplet must be accompanied by the evaporation of water in
order to maintain the equilibrium glycine concentration. A quantitative analysis of the Raman spectrum (Fig. 6b)
reveals that the spectral intensity of glycine modes with respect to the O-H stretching mode of water remains
constant, confirming a constant concentration of glycine molecules during the shrinking process. The measure-
ment ends when the particle becomes too small for stable trapping, and therefore leaves the optical trap.

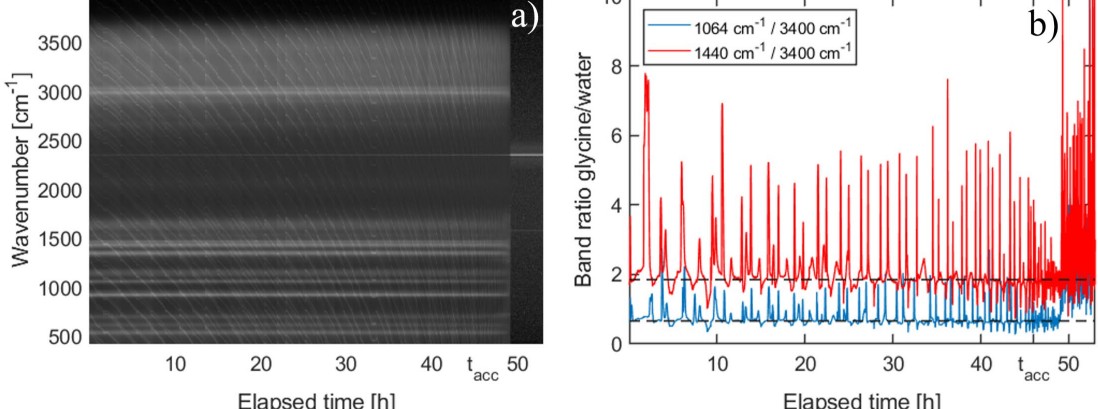

**Figure 6** Temporal evolution of Raman signal during droplet shrinking. **a)** Normalized Raman spectra stacked
chronologically from left to right. The molecular Raman bands are visible as horizontal straight white lines on the
dark background. The finer, slanted and curved lines correspond to whispering gallery modes. After approxi-
mately 49 h, the particle leaves the optical trap as it reaches a size that is too small for trapping, and only back-
ground is recorded. The band remaining afterwards at 2323 cm$^{-1}$ corresponds to the nitrogen gas in the trapping
cell. **b)** Ratio of the molecular glycine bands centered at 1064 cm$^{-1}$ and 1440 cm$^{-1}$ to the water band at around
3400 cm$^{-1}$. Peaks and dips in the graphs correspond to spectra where a whispering gallery mode is superimposed
on the glycine signal and the water signal, respectively, and are not relevant for the molecular composition. The
dashed horizontal lines are a guide to the eye. In this example, the acceleration of the droplet shrinking is observed
at $t_{acc}$ = 46 h. From this point in time onwards until the particle is lost, the faster shrinking leads to more frequent
WGMs perceived as an apparent increase of noise in the data.
The linear dependence of the shrinking rate on the light intensity (Fig. 5) proves that the observed shrinking is
induced by the laser light. This observation is intriguing as aqueous glycine is not known to absorb light in the
visible range, similar to other amino acids(Bhat and Dharmaprakash, 2002). Furthermore, the linear dependence
between shrinking rate and light intensity rules out the possibility of multiphoton absorption, which might other-
wise populate energetically excited states of glycine to induce chemical reactions.





The observation of a constant shrinking rate over a large portion of the experiments (Fig. 4) provides another
important piece of information, as this allows us to rule out some of the possible shrinking mechanisms. For
instance, assuming some residual absorption of light at 532 nm by the droplet solution, one might argue that the
shrinking is a consequence of enhanced evaporation due to the droplet heating by the laser. The evaporation of
glycine from the droplet can be approximated by the Hertz-Knudsen equation:
$$\frac{\mathrm{d}N}{\mathrm{d}t} = S \cdot \frac{\alpha p}{\sqrt{2\pi M R T}} \qquad (1)$$

where $\frac{\mathrm{d}N}{\mathrm{d}t}$ is the molar evaporation rate of glycine, $S$ is the droplet's surface, $p$ is the partial pressure and $M$ is the
molar mass of glycine, $R$ the gas constant, $T$ the temperature and $\alpha$ a heuristic sticking coefficient with values
between 0 and 1. It is evident from Eq. (1) that the rate of shrinking by evaporation should scale with the surface
area of the droplet, and that in this case a deceleration of the shrinking should be observed over time, contrary to
the experimental data. Moreover, any heating by laser light absorption would at most lead to a very small temper-
ature rise due to the efficient cooling by the gas flow. In addition to these arguments, if evaporation were signifi-
cant, some evaporation should still be observable even at low light intensity, where the heating of the droplet is
negligible and hence any deviations form room temperature can be neglected. As seen from Fig. 5 however, there
is no shrinking observable for low light intensities. We can therefore rule out evaporation as the dominant shrink-
ing mechanism.

The observation of a constant shrinking rate also excludes photochemical reactions in which molecular glycine
directly absorbs photons:
$$\mathrm{Gly(aq)} + h\nu \rightarrow \mathrm{P(aq)} \rightarrow \mathrm{P(g)}, \qquad (2)$$

where $\mathrm{Gly(aq)}$ is a solvated glycine molecule, $h\nu$ is the energy of the incoming photon and P is the reaction prod-
uct that is removed from the droplet (aq) into the surrounding gas phase (g) afterwards. As mentioned above.
molecular glycine is considered non-absorbing at 532 nm. However, if we nevertheless assume that molecular
glycine could be very weakly absorbing as in Eq. (2), the photon density inside the droplet would be constant over
the course of the photochemical reaction. Hence, the first step in Eq. (2) would be pseudo-first order, for which
the reaction rate is proportional to the concentration of glycine molecules in the droplet. Therefore, one would
expect a decrease in the observed shrinking rate over time, which contradicts the experimental observation.

The examples above illustrate that any mechanism, in which glycine molecules directly absorb incoming photons,
cannot explain the constant shrinking rates. This implies that a more intricate reaction must take place in the
droplet. We suggest the following simplified scheme with an additional reaction partner M:
$$\mathrm{M} + h\nu \rightarrow \mathrm{M}^* \qquad (3)$$

$$\mathrm{M}^* + \mathrm{Gly} \rightarrow \mathrm{M} + \mathrm{P} \qquad (4)$$



where $M^*$ denotes a photoexcited state of M, and P is the reaction product of glycine in the presence of this pho-
toexited species. This scheme represents a mediated reaction in which the reaction partner M is activated by light
absorption, and then reacts with glycine and returns back to the ground state. M shows the characteristics of a
photosensitizer (Corral Arroyo et al., 2018; George et al., 2015; Wang et al., 2020; Rapf and Vaida, 2016), which
is not consumed during the reaction, and therefore the amount of M remains constant in the droplet. We further
assume that the light absorption of the photosensitizer M is the rate limiting step, i.e., that Eq. (3) proceeds much
slower than Eq. (4). This is equivalent to requiring that M is only weakly absorbing, or that the concentration of
M in the droplet is low. Since the amount of M in the rate limiting step (Eq. (3)) remains constant during the
shrinking process, so does $M^*$ (quasistationary), resulting in a constant rate of degradation of Gly (Eq. (4)). As-
suming that the absolute concentration of M is much smaller than that of Gly at all times, and therefore has no
relevant effect on the equilibrium water vapor pressure of the droplet, the volume shrinking rate is predicted to be
constant in accordance with the experimental data.

Although the proposed mechanism in Eq. (3) and (4) concurs with the observed constant shrinking rates, it does
not yet specify the nature of the photosensitizer and its reaction with glycine. We first discuss potential candidates
for the photosensitizer. Contamination during the preparation of the different aqueous glycine solutions used in
this study was minimized by using pure substances (glycine purity ≥ 99 %, water resistivity 18.2 MΩ·s). No
correlation between the shrinking rates in Fig. 5 and the age of the solution at the time of the measurements was
observed, which indicates that there is no accumulation of photoactive contaminants in the solution after the
preparation. We therefore argue that contamination is not the origin of the photosensitizer.

It is evident from the previous discussion that the photosensitizer must possess an absorption band at 532 nm,
which is not the case for single solvated glycine molecules. The optical properties of molecules may change
however when forming intermolecular bonds, e.g. leading to an enhancement of the absorption and fluorescence
in the case of protein aggregates (Homchaudhuri and Swaminathan, 2004; Shukla et al., 2004; Chan et al., 2013;
Pinotsi et al., 2013) and amino acid clusters (Chen et al., 2018). For glycine solutions in particular, observations
of light absorption in the near UV/vis range have been attributed to the presence of mesoscopic clusters (Jawor-
Baczynska et al., 2013; Zimbitas et al., 2019), specifically due to the formation of hydrogen bonds between indi-
vidual molecules (Terpugov et al., 2021).

Mesoscopic clusters occur naturally in both undersaturated and supersaturated glycine solutions, though a kinetic
barrier may have to be overcome for their formation (Jawor-Baczynska et al., 2013). The average size of the
mesoscopic clusters, typically of the order of 100 nm for bulk solutions, depends not only on the monomer con-
centration but also on the history of the sample, indicating that the clusters do not necessarily remain in thermo-
dynamic equilibrium after formation (Jawor-Baczynska et al., 2013). Based on these observations, we propose
that the photo-induced droplet shrinking is mediated by light absorption of mesoclusters in the glycine solution.





These mesoclusters remain kinetically stable during the droplet shrinking, despite the decrease of the absolute
number of glycine monomers (i. e. constant glycine monomer concentration). Hence, these clusters act as stable
photosensitizers inside the droplets, in accordance with the observed constant shrinking rate. The observation of
a constant initial shrinking rate thus allows us to narrow down the possible reaction mechanisms at work in the
droplets. Alternative schemes might be conceivable, such as the existence of several reaction partners for glycine.
However, there is no further experimental evidence in favor of more complex alternatives to the simple mecha-
nism proposed (Eq. (3) and (4)). Furthermore, as pointed out above, mesoscopic glycine clusters match the req-
uisite characteristics of the proposed photosensitizer, and are therefore likely candidates. To pursue the argument,
let us further discuss the role of the mesoscopic clusters in the observed acceleration of the droplet shrinking.

The acceleration of the droplet shrinking proceeds relatively promptly at the point of acceleration when the par-
ticle has lost a typical amount of 75 % of its volume (Fig. 4). Assuming that the total number of mesoclusters
remains approximately constant (M in Eqs.(3) and (4)), their concentration has increased by an approximate factor
of 4 at this point. The sudden nature of the shrinking acceleration hints at a phase transition inside the particle. In
particular, the increase in nanocluster concentration may trigger the separation of a dense cluster phase inside the
droplets as part of a liquid-liquid phase transition. While a definite conclusion has to await more detailed micro-
scopic investigations, we present the following arguments in favor of this explanation. Mesoscopic clusters in
aqueous solutions are known to interact with focused light irradiation by assembling in the focal point of the light
beam due to the optical force that acts on the individual clusters (Sugiyama et al., 2012). This mechanism is the
basis of laser-induced phase transitions(Alexander and Camp, 2019), in the particular case of glycine both for
liquid-liquid phase separation (Sugiyama et al., 2012; Yuyama et al., 2010) and solid crystal nucleation (Sugiyama
et al., 2012; Alexander and Camp, 2019; Yuyama et al., 2010; Zaccaro et al., 2001; Garetz et al., 2002). It should
be noted at this point that the spot size in the focus of our optical trap is slightly larger (5 μm) than the typical
droplet size, and that therefore, it might appear unlikely that the electromagnetic field gradient is strong enough
to induce cluster aggregation in our case. However, micrometer sized droplets exhibit a large variance in the
spatial distribution of the internal light field, owing to the nanofocusing effect (Cremer et al., 2016; Corral Arroyo
et al., 2022), which can provide the field gradients necessary for aggregation. If in the case of our trapped droplets,
the acceleration is due to a phase transition, it would likely be assisted by the light irradiance. One would therefore
expect a dependence of the observed shrinking acceleration on the incident light intensity. Fig. 7 shows the meas-
ured acceleration ratio, that is, the ratio between the shrinking rate after to before the point of acceleration, as a
function of light intensity. From this data it is evident that the ratio increases with higher light intensity, which
agrees with our explanation of a light-induced phase transition. Since in this scheme, the clusters are expected to
aggregate in the regions of high light intensity after reaching a critical concentration, this would lead to a larger
absorption rate and thus a larger subsequent reaction rate, in agreement with the observation. Further studies will
be necessary to provide conclusive evidence for this explanation.





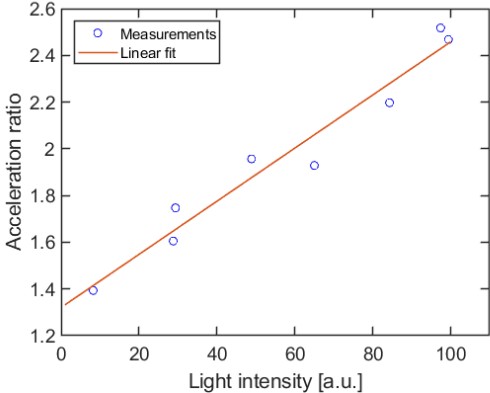


**Figure 7** Acceleration of droplet shrinking versus light intensity. The solid line represents the best linear fit
through the measurement data.


More data will also be required to understand the specifics of the interaction between the photosensitizer and the
solvated glycine molecules in Eq. (4). Here, we can only provide a qualitative discussion based on the available
data. The Raman spectra (Fig. 6) show no detectable change in the molecular composition, even after the droplet
has lost the major part of its volume during the shrinking process. Since this observation rules out the accumula-
tion of reaction products in the droplets over time, the reaction products must be small, volatile compounds that
quickly evaporate into the surrounding gas phase. Known degradation mechanisms of glycine and other amino
acids in aqueous solution proceed via reaction with radical species, in particular solvated electrons $e^-_{solv}$ and hy-
droxyl $^\bullet$OH radicals (Moenig et al., 1985; Garrison, 1964, 1972), which form as part of a photosensitized reaction
with chromophoric organic matter in water (Lundeen et al., 2014; Sun et al., 2018; Mopper and Zika, 1987).
Currently, our data does not allow us to distinguish between different degradation pathways.

## 4. CONCLUSIONS


We have demonstrated that single micrometer sized aqueous glycine droplets respond to the illumination with
laser light of 532 nm by shrinking, despite the fact that molecular glycine does not absorb in the near UV/vis
range. Most remarkably, the volume shrinking rate remained constant over the major part of the shrinking process.
This indicates a photo-induced decay of glycine molecules in the presence of a rate limiting catalyst, or photosen-
sitizer, and the subsequent evaporation of small, volatile reaction products. Based on the available literature data,
we propose that intrinsic mesoscopic clusters of glycine molecules formed by hydrogen bonding in the aqueous
solution are the most plausible candidates for this photosensitizer. The presence of mesoscopic glycine clusters
would also explain the sudden acceleration of the shrinking rate occurring at a volume loss of ~75 %. Because of
its dependence on the light intensity, we attribute this sudden rate change to arise from the interaction of the
mesoclusters with the incident light, possibly initiating a light-induced phase transition.



This study provides yet another example of the non-trivial interactions of light with aqueous glycine solutions (Alexander and Camp, 2019; Sugiyama et al., 2012; Zaccaro et al., 2001; Garetz et al., 2002; Yuyama et al., 2010), which facilitate previously undiscovered reaction pathways - interactions that are likely not exclusive to glycine. Light harvesting by and light interaction with such mesoscopic photosensitizers in aerosol droplets might also have played a role in the formation of more complex organic molecules under prebiotic conditions. Further investigations are needed to shed light on the specifics of the observed phenomena, and to yield new insight into the underlying reaction mechanisms, which remain elusive in part. Studying solutions in micrometer sized droplets (attoliter volumes) using high laser powers offers the advantage of much higher sensitivity to photo-induced reactions than typically achievable with bulk solutions.

**Acknowledgements**

This project was supported by the Japan Society for the Promotion of Science (JSPS Oversea Research Fellowship; S.I.), by the Swiss National Science Foundation (SNSF project number 200020_200306), and by ETH Zürich. We are very grateful to D. Stapfer and M. Steger from our workshops for technical support, and to D. Zindel for helpful discussions.

**Data Repository**

The data that support our findings are deposited on the ETH Research Collection under https://doi.org/xx.



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
