# Peer review of "Photo-Induced Shrinking of Aqueous Glycine Aerosol Droplets"

_Atmospheric Chemistry and Physics, 2023_

## Author Response (AR1)

**Referee 1's comments**

**Comment 0**
In this manuscript, Ishizuka et al. investigated the response of glycine droplets to visible light irradiation and proposed that the shrinkage of glycine droplets was due to the photosensitized oxidation of mesoscopic glycine clusters. However, I remain unconvinced of the validity of this proposed mechanism since the authors did not provide sufficient strong evidence to support it.

**Reply 0**
We very much appreciate the discussion about the proposed reaction mechanism. There are certain intriguing aspects that only future studies might be able to elucidate. The experimental data available from our study already leave a photochemical reaction as the only plausible mechanism for the observed phenomenon. In the following, we clarify some aspects of our arguments to address the comments by RC1. We suggest extending corresponding passages in the manuscript accordingly.

**Comment 1**
First of all, the authors have not conducted any control experiments that support the conclusion that the shrinkage of glycine droplets is induced by photochemistry. The authors could conduct experiments with non-photosensitive compounds to see the changes in droplet size over time. This would help confirm that the observed shrinkage is indeed caused by photochemistry.

**Reply 1**
The fact that the kinetics of the particle shrinking is proportional to the laser intensity with much slower shrinking rate at very low laser intensity is the unequivocal experimental proof that the light is driving the observed shrinking of the aqueous glycine droplets. As the light drives this shrinking, it can only be a photochemical reaction or induced by photothermal processes. As discussed in the manuscript, thermally driven evaporation can be ruled out by the observed shrinking dynamics, in particular, the constant volume shrinking rates. In addition, as explained in our Reply 3 to James Donaldson's comments, the temperature increase of the droplets is at most 0.7°C. Hence the shrinking we observe can only be a photochemical reaction. Nonetheless following the comment of the referee, we now provide measurements performed on non-photosensitive compounds (aqueous sea salt droplets) in a new Supplementary Material. These measurements show that non-photosensitive compounds do not experience such shrinking over 15 hours under exposure to a high intensity laser (2W laser power). This is a further proof that the observed shrinking is induced by photochemistry.

To address this question, we added the following text to the manuscript (replaces lines 166-170 in the old manuscript):
"Aqueous droplets which contain a stable non-volatile solute do not shrink over time when they are optically trapped (see Supplementary Section S1 for an example). The observation of droplet shrinking in the case of glycine as solute proves that glycine does not remain stable inside the droplet solution. Furthermore, the linear dependence of the shrinking rate on the light intensity (Fig. 5) proves that the observed shrinking is induced by the laser light. In particular, the volume shrinking rate increases from 0.18 to 2.25 $\mu m^3$ $h^{-1}$ when the light intensity is increased by the same relative amount, thus spanning over one order of magnitude. This observation is intriguing as aqueous glycine is not known to absorb light in the visible range, similar to other amino acids (Bhat and Dharmaprakash, 2002). In addition, the linear dependence between shrinking rate and light intensity rules out the possibility of multiphoton absorption, which might otherwise populate energetically excited states of glycine to induce

chemical reactions. Yet the fact that the observed shrinking is driven by the incident light indicates a photochemical reaction or photothermal processes occurring inside the droplet. As is shown in the following however, photothermal processes can be ruled out by our experimental data, leaving a photochemical mechanism as the only plausible option."

**Comment 2**
Besides, on line 195, the authors assumed that the photon density inside the droplet would be constant throughout the photochemical reaction. However, it is unclear whether this assumption is valid due to the optical confinement within the droplets, as discussed in a recent publication (Science 2022, 376:293). It is possible that the photon density within the droplets increases as their size decreases.

**Reply 2**
In some cases, optical confinement does induce a strong size dependence of the internal light intensity. However, for weakly absorbing particles such as aqueous glycine droplets, almost no size dependence is observed. Figure 4 of the publication mentioned by the referee (Science 2022, 376:293) shows that the internal light intensity in water droplets has almost no dependence on the particle radius between 200 nm and 5000 nm. Our work presented in Parmentier et al. *J. Aerosol Sci.*, **151**, 105660 (2021) also shows and discusses why no significant size dependence of the optical confinement is observed for weakly absorbing particles in the micrometer- and submicrometer-size ranges. Hence, there is no significant increase of the photon density in the studied droplets as their size decreases.

To clarify this question, we added the following sentence with two references in lines 207-209: "Earlier studies (Parmentier et al., 2021; Corral Arroyo et al., 2022) show that no significant size dependence of the optical confinement is observed for weakly absorbing particles in the micrometer- and submicrometer-size ranges."

**Comment 3**
On line 253, the authors state that the glycine concentration is constant (supersaturated) within the droplet, but the concentration of glycine clusters increases fourfold as the droplet loses 75% of its volume. This result is unexpected, as one might expect the concentration of glycine clusters to remain constant with the constant glycine concentration.

**Reply 3**
Indeed, in thermodynamic equilibrium, the concentration of glycine clusters would be coupled to the concentration of molecular glycine in the aqueous phase. However, as explained by Jawor-Baczynska et al. *Faraday Discuss.*, **167**, 425 (2013), glycine mesoscopic clusters have a formation barrier which also means a dissolution barrier. This barrier allows the clusters to be kinetically stable. Hence, even when the volume of the particle is divided by four, the glycine clusters, which are then four times more concentrated in the droplet, remain stable over the course of the measurements.

The text in lines 248-251 has been modified to clarify this point:
"Mesoscopic clusters occur naturally in both undersaturated and supersaturated glycine solutions, though a kinetic barrier has to be overcome for their formation (Jawor-Baczynska et al., 2013). A similar barrier needs to be overcome to dissolve these clusters. Hence the clusters can remain kinetically stable even if the number of glycine monomers decreases."

**Comment 4**

Lastly, since no products were detected in the droplet and it is unknown whether the mesoscopic glycine clusters are photosensitizers, further strong evidences are necessary to confirm that the observed shrinkage of the droplet is indeed caused by photochemical reactions.

**Reply 4**

As explained in our Reply 1 above, the intensity dependence of the reaction/shrinking rate unequivocally proves that the observed process is caused by a photochemical reaction. This is now discussed in more detail in the new paragraph in lines 170-181 of the manuscript (see above). Demonstrating that the mesoscopic clusters are the photosensitizer driving this photochemical reaction is more complicated and would require the determination of the internal structure of the droplets, e.g. by X-ray diffraction measurements. Theoretical approaches such as quantum molecular simulations of the glycine clusters are not feasible due to their size (1 nm to 100 nm; Hua et al., Cryst. Growth Des. **20**, 6502-6509 (2020)). This is beyond the scope of the present investigation. Nonetheless, we argue in the spirit of Occam's razor, that the mesoscopic clusters are the simplest and most likely candidates for a photosensitizer (the chemical analysis we performed of the aqueous glycine solution does not show any other possible candidates).

CC1 J.D. 20.022023

**Comment 0**
"This manuscript reports an observation of droplet shrinkage over time which, if the conclusions proposed by the authors are correct, would reprresent an exciting new finding. However, as presented in the present form, I am not convinced about the mechanism for shrinkage proposed by the authors."

**Reply 0**
Thank you very much for your comments. As we explain in more detail below, a photochemical reaction is the only plausible mechanism for the observed phenomenon. The mechanism proposed in your comment would be in contradiction to our experimental results. We take the opportunity to clarify a few aspects of our work and propose to elaborate some of the arguments in the manuscript accordingly.

**Comment 1**
"I would start an analysis from a different perspective: The Maxwell relation (derived from Fick's Law) predicts the rate of particle size change with time, Assuming that water (the highest volatility and greatest concentration component) is being lost to give rise to the shrinkage, the Maxwell equation predicts that the rate depends on {p*/T* - p#/T#), where T* and p* represent the temperature at the surface of the droplet, and the corresponding equilibrium water vapour pressure, respectively, and T#, p# represent those parameters in the ambient atmosphere - that is, the vapour pressure given by the 77% RH at the temperature of the background. It is known that T* = T# only if there is not evaporation taking place."

**Reply 1**
The analysis of the evaporation mentioned in the comment is applicable to pure substances, which are not in equilibrium with the surrounding gas phase. In our study, we trap aqueous glycine droplets at a RH of 77 %. For these droplets, the glycine dissolved in the droplet lowers the saturation vapor pressure of the droplet (as explained by Raoult's law), such that the water activity of the droplet matches the RH of the surrounding. Hence the water in the droplet is in thermodynamic equilibrium with that in the surrounding and does not evaporate, as long as the temperature of the droplet does not change and the amount of glycine in the droplet remains the same.

If, for sake of argument, we assumed that water does evaporate, one would expect the intensity of the water signature in the Raman spectrum (see for example the O-H stretching mode, 2700-3750 $cm^{-1}$) to decrease relative to the glycine signal during the shrinking process. We observe, however, a constant ratio between the Raman signal of glycine and water during the experiments. Evaporation of water from the droplet's surface would be expected to

decelerate in terms of volume shrinking rate (see Eq. 1 and the discussion in line 183-198 in the main text). This is not observed in the experiment.

An additional argument, why evaporation of water alone cannot explain the droplet shrinking, is the fact that water only makes up approximately 40 % of the droplet mass (see also line 62). Using the densities of pure water and pure glycine to approximately convert mass to volume, the initial water volume corresponds to ca. 50 % of the initial particle volume. However, our experiments show (Fig. 4) that for some particles, more than 90 % of the volume is lost during the experiments. Notably, even at this point, the glycine-to-water ratio does not change, as indicated by the Raman spectra.

**Comment 2**
"In the absence of the laser, one would expect the droplet size to change as given above."

**Reply 2**
We would like to point out that no shrinking of the droplet is observed in absence of the laser (i.e. when we extrapolate the data in Fig. 5 towards zero light intensity). As explained in Reply 1, this is due to the fact that without light irradiation, glycine remains stable in the droplet solution.

**Comment 3**
There is a small absorption of 532 nm light by liquid water. Although very weak, it is present; one can claculate a 10s (or more!) of degrees temperatute rise in the droplet **over the 10s-100s of hours of the experiment;** this would certainly change the evaporation. The authors state that the heat transfer to the surrounding gas maintains a constant droplet temperature, but do not test this assertion.

Another effect arises from the Raan pumping of the droplet components. Again, the energy of the vibrationally excited species is expected to be small (but can be estimated), but over 10s to 100s of hours may also give rise to a temperature increase in the droplet which is not considered here.

**Reply 3**
Water has indeed some residual light absorption in the visible range. Therefore, a finite temperature increase of the droplet with respect to its surrounding is expected. The predicted effect is, however, not a continuous increase of temperature as suggested. Instead, the temperature will equilibrate at some value above room temperature where the heat flux from the droplet to the surrounding is equal to the light absorption rate. Assuming that the aqueous glycine droplet has a similar absorption rate as a droplet consisting of pure water (complex refractive index water: $1.334 + i \cdot 1.5 \cdot 10^{-9}$), we estimate the temperature of the droplet to increase by at most 0.7 °C, which is negligible in our experiments.

Let us assume, for the sake of argument, that heating were significant in our experiment and that the droplet would equilibrate at a significantly elevated temperature. The water vapor pressure of the droplet would inevitably increase, which would lead to evaporation of water and shrinking of the droplet. As a consequence, the glycine concentration would increase, until the water vapor pressure of the droplet would again be equal to the partial pressure of water in the surrounding (RH ca. 77 % at T = 21°C).  At this point, the droplet would be in equilibrium again, and the shrinking would stop. This predicted behavior does not reflect our experimental data, which clearly shows that the droplet shrinks continuously, with no deceleration of the shrinking rate observable. Also, as explained in Reply 1, the glycine concentration is not observed to increase at any given time.

These considerations show that for continuous shrinking of the droplet, glycine has to be removed from it. Any mechanism, where only the water component of the droplet evaporates, cannot explain the experimental data.

As both this comment and the comment by Referee #1 raise questions about the proposed shrinking mechanism, we extend the paragraph in line 170-181 to clarify why only a photochemical reaction can explain our data. The new paragraph reads as follows:

"Aqueous droplets which contain a stable non-volatile solute do not shrink over time when they are optically trapped (see Supplementary Section S1 for an example). The observation of droplet shrinking in the case of dissolved glycine proves that glycine does not remain stable inside the droplet solution. Furthermore, the linear dependence of the shrinking rate on the light intensity (Fig. 5) proves that the observed shrinking is induced by the laser light. In particular, the volume shrinking rate increases from 0.18 to 2.25 $\mu m^3$ $h^{-1}$ when the light intensity is increased by the same relative amount, thus spanning over one order of magnitude. This observation is intriguing as aqueous glycine is not known to absorb light in the visible range, similar to other amino acids (Bhat and Dharmaprakash, 2002). In addition, the linear dependence between shrinking rate and light intensity rules out the possibility of multiphoton absorption, which might otherwise populate energetically excited states of glycine to induce chemical reactions. The fact that the observed shrinking is driven by the incident light indicates a photochemical reaction or photothermal processes occurring in the aqueous glycine droplet. As is shown in the following however, photothermal processes can be ruled out by our experimental data."

We also add a supplementary section to show the behavior of a droplet that contains a photochemically inert solute for comparison with aqueous glycine droplets, and one section to show the calculation that leads to the estimated maximum of 0.7 °C for the temperature elevation of the droplet with respect to the surrounding.

Comment 4

A smaller point is that "99% glycene" is, by definition 1% somethig else and that 18 MOhm water is generally not necessarily free of organic chromophores."

Reply 4

The manufacturer of the glycine used for preparation of the solutions (Sigma-Aldrich G7126) states that no contamination is detectable in their product. The Certificate of Analysis provided by the manufacturer for the particular lot we used specifies a purity of 100% determined by HPLC. We checked that the UV spectra of the solutions do not show any impurities. In response to this comment, we also performed a chemical analysis of the aqueous glycine solutions from which the droplets were generated. We have performed an elementary analysis and an HPLC analysis of the glycine as provided by the manufacturer and of our glycine solution. The elementary analysis showed no indication of any impurities. From this, we deduce a purity of at least 99.9%. The HPLC of the glycine (measured as N-trifluoroacetyl-glycine-isopropyl ester) did not give any indication of a detectable impurity that could absorb visible light (wavelength 532 nm). Given the available data, we argue that it is highly unlikely that our samples contain a contamination that could act as a photosensitizer in our sample to any significant extent.

To clarify this, we have added the following text in lines 51-56:
"The droplets are generated from 1.0 M or 2.0 M aqueous solutions of glycine (purity 100%, HPLC certificate, Sigma-Aldrich G7126) using a commercial atomizer (TSI 3076) with pressurized, humidified nitrogen gas (purity 5.0). We performed additional elementary and HPLC analyses of the glycine and the glycine solutions. The results showed no indication of any trace impurities absorbing in the visible range. On this basis, we conclude that our samples do not contain any contaminants that could act as photosensitizer for photochemical reactions (see below)."